# Evaluation of CMIP6 model simulations of PM$_{2.5}$ and its components over China

Fangxuan Ren[1], Jintai Lin[1], Chenghao Xu[1], Jamiu A. Adeniran[1], Jingxu Wang[2], Randall V. Martin[3], Aaron van Donkelaar[3], Melanie S. Hammer[4], Larry W. Horowitz[5], Steven T. Turnock[6,7], Naga Oshima[8], Jie Zhang[9], Susanne Bauer[10], Kostas Tsigaridis[11,10], Øyvind Seland[12], Pierre Nabat[13], David Neubauer[14], Gary Strand[15], Twan van Noije[16], Philippe Le Sager[16], Toshihiko Takemura[17]

[1] Laboratory for Climate and Ocean-Atmosphere Studies, Department of Atmospheric and Oceanic Sciences, School of Physics, Peking University, Beijing 100871, China

[2] Frontier Science Center for Deep Ocean Multispheres and Earth System (FDOMES) and Physical Oceanography Laboratory, College of Oceanic and Atmospheric Sciences, Ocean University of China, Qingdao 266100, China

[3] Department of Energy, Environmental, and Chemical Engineering, Washington University, St. Louis, MO, USA

[4] St. Francis Xavier University, Department of Earth Sciences, Antigonish, NS, Canada

[5] NOAA Geophysical Fluid Dynamics Laboratory, Princeton, NJ, USA

[6] Met Office Hadley Center, Exeter, UK

[7] University of Leeds Met Office Strategic (LUMOS) Research Group, University of Leeds, UK

[8] Meteorological Research Institute, Tsukuba, Japan

[9] Beijing Climate Center, China Meteorological Administration, Beijing 100081, China

[10] NASA Goddard Institute for Space Studies, New York, NY, USA

[11] Center for Climate Systems Research, Columbia University, New York, NY, USA

[12] Norwegian Meteorological Institute, P.O. Box 43 Blindern, Oslo, Norway

[13] Centre National de Recherches Météorologiques (CNRM), Météo-France, CNRS, Toulouse, France

[14] Institute of Atmospheric and Climate Science, ETH Zurich, Zurich, Switzerland

[15] Climate and Global Dynamics Laboratory, the National Center for Atmospheric Research, Boulder, CO, USA

[16] Royal Netherlands Meteorological Institute, De Bilt, Netherlands

[17] Research Institute for Applied Mechanics, Kyushu University, Fukuoka, Japan

*Correspondence to:* Jintai Lin (linjt@pku.edu.cn)

**Abstract.** Earth system models (ESMs) participating in the latest Coupled Model Intercomparison Project Phase 6 (CMIP6) simulate various components of fine particulate matter (PM$_{2.5}$) as major climate forcers. Yet the model performance for PM$_{2.5}$ components remains little evaluated due in part to lack of observational data. Here, we evaluate near-surface concentrations of PM$_{2.5}$ and its five main components over China as simulated by fourteen CMIP6 models, including organic carbon (OC, available in 14 models), black carbon (BC, 14 models), sulfate (14 models), nitrate (4 models), and ammonium (5 models). For this purpose, we collect observational data between 2000 and 2014 from a satellite-based

dataset for total $PM_{2.5}$ and from 2469 measurement records in the literature for $PM_{2.5}$ components. Seven models output total $PM_{2.5}$ concentrations, and they all underestimate the observed total $PM_{2.5}$ over eastern China, with GFDL-ESM4 (–1.5%) and MPI-ESM-1-2-HAM (–1.1%) exhibiting the smallest biases averaged over the whole country. The other seven models, for which we recalculate total $PM_{2.5}$ from the available components output, underestimate the total $PM_{2.5}$ concentrations, partly because of the missing model representations of nitrate and ammonium. Concentrations of the five individual components are underestimated in almost all models, except that sulfate is overestimated in MPI-ESM-1-2-HAM by 12.6% and in MRI-ESM2-0 by 24.5%. The underestimation is the largest for OC (by –71.2% to –37.8% across the 14 models) and the smallest for BC (–47.9% to –12.1%). The multi-model mean (MMM) reproduces fairly well the observed spatial pattern for OC (R = 0.51), sulfate (R = 0.57), nitrate (R = 0.70) and ammonium (R = 0.74), yet the agreement is poorer for BC (R = 0.39). The varying performances of ESMs on total $PM_{2.5}$ and its components have important implications for the modeled magnitude and spatial pattern of aerosol radiative forcing.

**1 Introduction**

Fine particulate matter ($PM_{2.5}$) influences air quality, human health and climate change. Exposure to near-surface $PM_{2.5}$ is associated with millions of global premature deaths each year (Zhang et al., 2017; World Health Organization, 2021). $PM_{2.5}$ affects the radiative budget of the climate system directly through scattering and absorption and indirectly via clouds. The effects of atmospheric aerosols on cloud droplet concentrations, cloud distributions and radiative properties pose large uncertainties in the estimating radiative forcing (Carslaw et al., 2013; Seinfeld et al., 2016). Earth system models (ESMs) are essential tools for studying global climate change. The accuracy of $PM_{2.5}$ simulations in ESMs exhibits a crucial constraint on the reliability of these models in climate change simulation and projection. The Coupled Model Intercomparison Project Phase 6 (CMIP6) provides an opportunity to evaluate simulated $PM_{2.5}$ and its components by the current-generation ESMs, which implement interactive aerosol and atmospheric chemistry (Turnock et al., 2020). A total of 21 ESMs participating in CMIP6 provide total $PM_{2.5}$ and/or several component simulations, although the aerosol component species vary across these models. Fourteen models include organic aerosol (OA, converted to organic carbon (OC) in this study by assuming OA / OC = 1.6), black carbon (BC), sulfate, dust (DST), and sea salt (SSLT). Four of these

14 models also include nitrate and five include ammonium (Table S1).
Aerosol optical depth (AOD) during 2000–2014 simulated in CMIP5 and CMIP6 are in broad agreement
with satellite retrievals over most parts of Europe, North America, and India (Zhang et al., 2022a; Cherian
and Quaas, 2020). CMIP6 models better capture satellite-based AOD trends in western North America
and eastern China, whereas CMIP5 models failed to reproduce the trends in AOD (Mortier et al., 2020;
Cherian and Quaas, 2020). Studies have emerged over recent years to assess the CMIP model
performance of individual aerosol components. An assessment of CMIP5 dust aerosol simulations using
independent data from 1851 to 2011 over North Africa shows a common underestimate (Evan et al.,
2016). Another analysis of CMIP3 and CMIP5 models suggests sea salt aerosols over the tropical Pacific
to be significantly underestimated (Chen et al., 2020). Evaluation of the vertical distribution of BC in
CMIP5 models based on aircraft measurements shows an overestimate in the upper troposphere
especially over the Central Pacific (Allen and Landuyt, 2014). Several CMIP5 models produce high
sulfate burdens over eastern China, the Indian Peninsula and the northern Indo-Chinese Peninsula,
although the transport difference among these models results in distinctive spatial distributions (Li et al.,
2020). Overall, global climate models struggle to accurately reproduce observed aerosol component
concentrations over different world regions.
China is a major region with heavy aerosol pollution, dense population and complex climate, and thus it
is critical to understand the performance of ESMs for aerosol simulations over this country. Several
studies have evaluated total $PM_{2.5}$ simulations of CMIP models over China, using AOD data from
satellite retrievals (Sockol and Small Griswold, 2017; Michou et al., 2020) and ground-based aerosol
networks (Mortier et al., 2020). They find that CMIP5 models reproduce the spatial pattern of AOD
reasonably well over eastern China, but with a tendency to underestimate AOD magnitudes (Liu and
Liao, 2017; Park et al., 2014; Allen et al., 2013). GFDL-CM3 performs best among CMIP5 models in
simulating AOD over eastern China, partly because it includes nitrate and ammonium that most models
lack (Li et al., 2020). Other studies suggest that CMIP6 models simulate the magnitude of annual mean
AOD better than CMIP5 over eastern China, in part due to the notable increase in sulfate (Cherian and
Quaas, 2020; Fan et al., 2018a). Nonetheless, the CMIP6 models fail to capture the seasonal north-south
shift of AOD maximum center over China during 2000–2014 (Li et al., 2021) and the observed dipole
pattern of AOD trends between China and India during 2006–2014 (Wang et al., 2021b).
Different $PM_{2.5}$ components exhibit distinctive radiative effects, thus understanding the performance of
ESMs in simulating individual $PM_{2.5}$ components is important. Due to the absence of publicly available
observational component data over China, only a few studies target single aerosol components (such as
sulfate and dust) over a large region of the country, or different $PM_{2.5}$ components over a short period or
a small region (Pu and Ginoux, 2018; Zhao et al., 2022). For example, model evaluation based on the
Acid Deposition Monitoring Network in East Asia (EANET) suggests that sulfate concentrations
simulated by CMIP5 and CMIP6 show a rising trend similar to observations (Mulcahy et al., 2020), but
the simulations are still lower than observed concentrations (Fan et al., 2018b; Mortier et al., 2020). A
recent study compares $PM_{2.5}$ components (dust, sea salt, BC, OC and sulfate) in CMIP6 models with the
Modern Era Retrospective analysis for Research and Applications Aerosol Reanalysis (MERRAero) in
Asia from 2005 to 2020 (Su et al., 2022; Buchard et al., 2016). The study shows that CMIP6 model
uncertainties of total $PM_{2.5}$ over East Asia are mainly attributable to sulfate and mineral dust simulations.
However, the model biases may in part come from other components (nitrate and ammonium) that are
not analyzed in their study; and the MERRAero data might contain errors as well (Ma et al., 2021;
Mahesh et al., 2019).
In this study, we evaluate near-surface concentrations of $PM_{2.5}$ and its five main components (OC, BC,
sulfate, nitrate, and ammonium) from 2000 to 2014 over China simulated by fourteen CMIP6 models
driven by historical emissions. For this purpose, we employ a satellite-based dataset for total $PM_{2.5}$
concentrations and a self-compiled $PM_{2.5}$ component dataset from 221 ground stations during 2000–2014
collected from the literature. Section 2 introduces CMIP6 model simulations, satellite-based total $PM_{2.5}$
concentration data, and literature-based $PM_{2.5}$ component data. Section 3 assesses the performance of
CMIP6 models for total $PM_{2.5}$. Section 4 evaluates the simulated $PM_{2.5}$ components. Section 5 discusses
the climate implications of the inadequacies in total $PM_{2.5}$ and its components in CMIP6 models. Section
6 concludes the study.
**2 Data and method**
**2.1 CMIP6 simulations**
Near-surface concentrations of total $PM_{2.5}$ and its components can be converted from dry aerosol mass
mixing ratios (MMRs) in CMIP6 models. Monthly mean near-surface MMRs (in the lowest model layer)
of $PM_{2.5}$ and its main components are taken from fourteen CMIP6 models to assess the performance of
ESMs over China (Table S1). Data are obtained from the "Historical" experiments covering 1850−2014,
which serve as the entry cards for participating in CMIP6 (Eyring et al., 2016). They are coupled
atmosphere-ocean simulations that include all CMIP6 historical forcings, and are well suited for
quantifying and understanding model characteristics. The ensemble mean is taken for each model by
averaging all available ensemble members. For GISS models, the ensemble members use two physics
configurations with drastically different aerosol parameterizations. We average the ensemble members
using the same physics configurations in GISS models, named GISS-E2-1-OMA (physics-version = 3)
and GISS-E2-1-MATRIX (physics-version = 5) respectively (Bauer et al., 2020). Simulation results over
2000−2014 are selected and re-gridded to $1° \times 1°$ for comparison with available satellite- and ground-
based data.
The anthropogenic emission data (ver. 2016-07-26) to drive "Historical" CMIP6 simulations is produced
by the Community Emissions Data System (CEDS) (Hoesly et al., 2018). An updated version of CEDS
(ver. 2017-05-18) corrected several errors in the spatial distribution within each country, but does not
change total emissions by country and sector (Feng et al., 2020). The CEDS emissions (ver. 2016-07-26
and ver. 2017-05-18) of OC, BC, CO, $NO_x$ and $SO_2$ in China after 2000 are higher than those in the
Multi-resolution Emission Inventory for China (i.e., MEIC) (Paulot et al., 2018; Zheng et al., 2018) and
the Peking University (PKU) inventory (Wang et al., 2014; Huang et al., 2015; Tao et al., 2018) which
use more detailed Chinese data. This difference in China has been reduced when CEDS was used to
derive future SSP scenarios in CMIP6 simulations (published on ESGF on 28 June 2018 on https://esgf-
node.llnl.gov/search/cmip6), and has been included in a post-CMIP6 version of CEDS (McDuffie et al.,

144 2020).

Of the fourteen models, all output the MMRs of OA, BC, sulfate, dust and sea salt, five output ammonium,
and four output nitrate (Table S1). Seven models output the MMRs of total $PM_{2.5}$, as the sum over all
components with suitable particle sizes. The MMRs are converted to mass concentrations ($\mu g\ m^{-3}$) based
on air density in each model. In evaluating $PM_{2.5}$ components (Sect. 4), the evaluation of dust and sea
salt concentrations is excluded due to the lack of available ground-based observations. We compare OC,
BC, sulfate, nitrate, and ammonium simulations with the observed data available for these components.
Modeled OA is converted to organic carbon (OC) to be comparable with the observational dataset.
Modeled OA refers to total organic aerosol, including primary organic aerosol (POA) and secondary
organic aerosol (SOA). For the GFDL-ESM4 model, the "mmroa" variable for OA only includes POA;
thus we calculate the total OA of GFDL-ESM4 as mmroa plus mmrsoa. The OA/OC ratios in the
literature range from 1.4 to 2.1 (Bürki et al., 2020; Lin et al., 2016). We choose an OA/OC ratio of 1.6,
which is the same as the ratio used in converting near-surface OA observations to OC. This ratio is
slightly higher than the value of 1.4 recommended by CMIP6 for POA, but it does not affect the relative
(percentage) model bias found in this study because the same ratio is used for models and observations.
For the seven models that do not output total $PM_{2.5}$, we follow the previous work to estimate total $PM_{2.5}$
concentrations (Eq. 1) (Turnock et al., 2020). Here, OA, BC, sulfate and certain portions of sea salt (SSLT,
$a_1$) and dust (DST, $a_2$) are assumed to be present in fine particles (diameter < 2.5 μm).
$$PM_{2.5} = OA + BC + SO_4^{2-} + a_1 \times SSLT + a_2 \times DST \tag{1}$$
For most models, specific values of $a_1$ and $a_2$ are provided by model developers (Table S2). BCC-ESM1
does not provide the coefficients. Instead, the model outputs concentrations in four size bins for each of
dust (DST01: 0.1−1.0 μm, DST02: 1.0−2.5 μm, DST03: 2.5−5.0 μm, and DST04: 5.0−10 μm) and sea
salt (SSLT01: 0.2−1.0 μm, SSLT02: 1.0−3.0 μm, SSLT03: 3.0−10 μm, and SSLT04: 10−20 μm) (Su et
al., 2022; Wu et al., 2019). Thus, the first two bins are assumed to belong to $PM_{2.5}$. Ammonium and
nitrate are not available in most of these six models (except GISS-E2-1-MATRIX) and are thus not
included in Eq.1.
**2.2 Satellite-based total $PM_{2.5}$**
We take satellite-based near-surface total $PM_{2.5}$ concentrations from the V4.CH.03 product of the
Washington University Atmospheric Composition Analysis Group (Hammer et al., 2020). The dataset is
constructed by combining multiple satellite products of AOD with simulations from a chemical transport
model (GEOS-Chem) to predict $PM_{2.5}$, and then constraining these estimates by ground-level $PM_{2.5}$
monitoring. The GEOS-Chem aerosol simulations include primary and secondary carbonaceous aerosols,
sulfate, nitrate, ammonium, mineral dust, and sea salt. The dataset provides the annual average $PM_{2.5}$
concentrations during the period 2000–2014 with a high spatial resolution of $0.01° × 0.01°$ (~$1 × 1$ $km^2$).
The adjusted satellite-derived $PM_{2.5}$ concentrations over Asia are compared with surface $PM_{2.5}$
observations collected from the Global Burden of Disease (GBD) collaborators during the period 2008–
2013 ($Mean_{satellite}$ = 61.5 µg $m^{-3}$ versus $Mean_{obs}$ = 59.1 µg $m^{-3}$) (van Donkelaar et al., 2016) and from the
China National Environmental Monitoring Center (CNEMC) during the period 2015–2019 ($Mean_{satellite}$
= 45.9 µg $m^{-3}$ versus $Mean_{obs}$ = 43.4 µg $m^{-3}$) (van Donkelaar et al., 2021). Detailed data descriptions are
provided elsewhere (van Donkelaar et al., 2019; van Donkelaar et al., 2016). Here the satellite-based
total $PM_{2.5}$ data are re-gridded to $1° × 1°$ for model evaluation purposes.
**2.3 Ground-based $PM_{2.5}$ components data**
Since national-scale continuous measurements of near-surface $PM_{2.5}$ components are unavailable in
China, we collect observational $PM_{2.5}$ component data from the literature. Our collected dataset includes
2469 component records of OC, BC, sulfate, nitrate, and ammonium nationwide (627, 66, 645, and 1131
records in western regions, Northeast China, North China, and Central and South China, respectively),
as shown in Figure 1. Here a record represents one measured $PM_{2.5}$ component at the specific sample site
and period. These records cover 30 provinces (including provinces and provincial-level municipalities)
and multiple land use types (urban, rural, near the road, and industrial park, etc.). The dataset does not
cover Ningxia, Guizhou, Heilongjiang, and Taiwan. A total of 472, 459, 518, 519, and 501 records are
available for OC, BC, sulfate, nitrate, and ammonium over China, respectively. The site locations,
sampling periods, data sources, and other information are summarized in the Supplement.
At a given site, the records are not continuous in time. These records cover varying sampling periods
ranging from a few days to several years, although most are monthly data. We treat a record as seasonal
if its data length is equal to or shorter than a season, or as annual when its data length is longer than 6
months. The records are not evenly scattered across years and are more available in later years in general.
From 2000 to 2008, the numbers of records range from 50 to 150 per year, except for 2003 (207 records);
while from 2009 to 2014, the numbers of records vary between 150 to 550 per year (Fig. S1). To compare
with CMIP6 simulations, we calculate for each site the multi-year mean PM$_{2.5}$ component concentrations
by averaging over the seasonal or annual observational records. If there are more than one sites in a given
model grid cell, we average data from all sites in that grid cell. To consider the effect of interannual
variability (caused by incomplete temporal match in data availability between models and observations),
we compute for each CMIP6 model the average and maximum of annual mean values during 2000–2014
from all grid cells with available observational data, and then compare with the multi-year averaged
observations from these grid cells. As detailed in Section 5, the model biases are not caused by imperfect
model-observation matching in time.
**3 Evaluation of near-surface total PM$_{2.5}$**
**3.1 Spatial distribution**
The spatial distribution of satellite-based annual mean total PM$_{2.5}$ concentrations (Fig. 2 p) exhibits high
values over populous and industrial North China (including Beijing, Tianjin, Hebei, Shandong, and
Shanxi provinces, 52.6 μg m$^{-3}$) and eastern Sichuan (60.9 μg m$^{-3}$). Central and South China exhibits
PM$_{2.5}$ concentrations (46.5 μg m$^{-3}$) lower than North China, due to lower emissions, higher vegetation
coverage, better ventilation conditions and more precipitation. PM$_{2.5}$ concentrations are modest over
dusty southern Xinjiang (33.6 μg m$^{-3}$). Low PM$_{2.5}$ concentrations (< 8 μg m$^{-3}$) are distributed over the
plateaus or forested regions with small populations, such as Tibet and northern Heilongjiang. Overall,
PM$_{2.5}$ concentrations in the south and coastal regions are lower than in the northern and inland regions.
Among the seven models that directly output total PM$_{2.5}$ concentrations (Fig. 2 a-g), GFDL-ESM4 and
MPI-ESM-1-2-HAM show similar patterns and magnitudes to satellite data with small national average
biases (–1.5% and –1.1%, respectively) because of better performance in BC, sulfate, and ammonium
simulations (Fig. S4-S7), which are related to the aerosol-chemistry-climate schemes within CMIP6
models (Turnock et al., 2020). Over the eastern regions (including Northeast China, North China, and
Central and South China), all models exhibit spatially averaged negative biases ranging from by –47.9%
to –3.3% (Fig. S2). Nevertheless, the spatial pattern over the eastern regions is well simulated by four
models (GFDL-ESM4, GISS-E2-1-OMA, MIROC-ES2L, and MPI-ESM-1-2-HAM) (R > 0.9, as shown
in Table S2) with the maximum center over North China correctly reproduced. Over the western regions,
four models (GFDL-ESM4, MRI-ESM2-0, NorESM2-LM, and NorESM2-MM) reproduce the
maximum center over southern Xinjiang, although each of the seven models can underestimate or
overestimate the peak values substantially.
For the seven models with total $PM_{2.5}$ derived from Eq.1, their simulated $PM_{2.5}$ concentrations
underestimate the satellite-based data by –65.5% to –48.0% averaged over the country (Fig.2 h-n). The
negative biases are in part because nitrate and ammonium are not included. About 15.1–20.6% and 11.4–
14.6% of $PM_{2.5}$ are nitrate and ammonium in the models that do contain them, as shown in Table S3.
Over the eastern regions, HadGEM3-GC31-LL and UKESM1-0-LL exhibit the least underestimation,
and they also capture the observed maximum center over North China. Five of these seven models do
not reproduce the $PM_{2.5}$ peaks over dusty regions in the west, pointing to model deficiencies in dust
simulations (Zhao et al., 2022).
**3.2 Trend and interannual variability**
Over the eastern regions (Northeast China, North China, and Central and South China), data from satellite
(0.72 μg m$^{-3}$ yr$^{-1}$) and all models (0.32–1.14 μg m$^{-3}$ yr$^{-1}$) exhibit significant increases (*p*-value < 0.05)
in annual mean total $PM_{2.5}$ concentrations over 2000–2014, with temporal correlation between 0.63 and
0.87 (Fig. 3 a and Table S2). The positive trend of satellite data over the eastern regions is consistent
with findings from previous studies (de Leeuw et al., 2022; Geng et al., 2021) , as caused mainly by
emission changes (Hoesly et al., 2018; Wang et al., 2022). GFDL-ESM4 and MPI-ESM1-2-HAM exhibit
annual average $PM_{2.5}$ concentrations and trends similar to the satellite data since 2004. Regionally, the
fourteen models capture the interannual variations of satellite $PM_{2.5}$ over Northeast China (R > 0.9) and
North China (R > 0.8) (Fig. 4). The temporal consistency reflects that the models capture the temporal
changes in anthropogenic emissions over these polluted regions, although the models might not align
with natural (meteorology-driven) variability.
Over the western regions where natural dust dominates the aerosol loadings, satellite-based $PM_{2.5}$
concentrations experience no significant trend over 2000–2014, whereas 11 models increase significantly
ranging from 0.10–0.28 μg m$^{-3}$ yr$^{-1}$) (Fig. 3 b). There is a notable decline over 2000–2005 in satellite
data (–1.12 μg m$^{-3}$ yr$^{-1}$, at the significance level of 0.1) consistent with the previous studies that use dust
aerosol optical depth (DOD) and ground-based observations of dust storm (Wang et al., 2021a; Song et
al., 2016). However, the dramatic drop is not captured by any model, reflecting large uncertainties and
inter-model diversities in dust simulations stemming from many factors such as the driving mechanisms,
dust particle size, and model structural differences (Zhao et al., 2022). Over 2000–2014, NorESM2-LM,
NorESM2-MM, and MPI-ESM-1-2-HAM show large interannual variations whereas other models do
not. The models do not align with the yearly changes found in the satellite data, with modestly positive,
low or even negative correlation coefficients (–0.6 to 0.6, Fig. 4). The inaccuracy in aerosol trend and
variability might exert erroneous forcing upon the climate system.
**4 Evaluation of near-surface PM$_{2.5}$ components**
**4.1 Organic carbon and black carbon**
Ground-based observations of carbonaceous aerosols (OC and BC) are mostly available in the eastern
regions. The national average multi-year mean observed OC concentration reaches 15.9 μg m$^{-3}$.
Observed OC concentrations peak over North China (> 25 μg m$^{-3}$) and are also high over Central and
South China (5–25 μg m$^{-3}$) (Fig. 5 a). The national average of the 14-model mean (6.5 μg m$^{-3}$, normalized
mean bias (NMB) = –59.0%), which is spatially coincidently sampled with the ground-based
observations (i.e., model values are obtained from grid cells with available observations), severely
underestimates the observations, especially over parts of North China with the bias reaching –40 μg m$^{-3}$
(Fig. 5 b). Nevertheless, the spatial pattern of OC observations is captured by the 14-model mean
modestly well with a correlation coefficient of 0.51. Further, a negative bias exceeding –50% occurs in
11 models, even though they can simulate the spatial pattern moderately well (R ranges from 0.40 to
0.58, Fig. S4).
The national average multi-year mean observed BC concentration is 4.3 μg m$^{-3}$. Observed BC
concentrations are high (> 10 μg m$^{-3}$) over parts of North China with mining and other heavy industries,
such as Hebei and Shanxi province (Fig. 5 d). However, the 14-model mean (3 μg m$^{-3}$) does not capture
the spatial pattern very well (R = 0.39) and it underestimates the observations (NMB = –27.2%). The 14-
model mean presents the largest negative bias over Shanxi (–15.2 μg m$^{-3}$) and the greatest positive bias
over Shandong (3.9 μg m$^{-3}$, Fig. 5 e); both provinces are in North China. Twelve of the 14 models
underestimate the BC observations (by –47.9% to –12.1% for national average), whereas two models
(HadGEM3-GC31-LL and UKESM1-0-LL) exhibit positive biases (by 21.1% and 26.2%, respectively)
(Fig. 6 and Fig. S5). Most models produce high concentrations of BC over the whole North China,
including Beijing and Shandong that exhibit relatively low observational values. The spatial distributions
of carbonaceous aerosol concentrations are mainly influenced by CEDS emissions used in models, with
higher spatial correlation coefficients greater than 0.85 (Fig. S3).
The underestimation of carbonaceous aerosol concentrations might be associated with anthropogenic
emissions, chemical mechanisms, and meteorological conditions. For China, the CEDS emission data
(ver. 2016-07-26) used in CMIP6 historical simulations are about 3.8–31.3% higher than those in MEIC
inventory except for $NO_x$ emissions (Fan et al., 2022). However, the positive bias in emissions cannot
explain the model underestimation of OC and BC concentrations. The model inadequacies in chemical
processes (e.g., using simplified aerosols and chemistry schemes, which tends to underestimate aerosol
formation (Turnock et al., 2020)) might lead to underestimated secondary organic aerosols (SOA, as a
component of OC), especially over Central and South China (Chen et al., 2016). The inter-model
discrepancies of OC and BC peak over North China and eastern Sichuan (Fig. 5 c). The large absolute
discrepancies are in part due to the higher air pollutant concentrations in these regions. Furthermore,
many differences exist among CMIP6 models in $PM_{2.5}$ component simulations, including the
representation of aerosol size distribution; the simplification of chemical processes with photolytic,
kinetic and heterogeneous reactions (e.g., 33 photolytic reactions in BCC-ESM1 but 43 in GFDL-ESM4)
(Turnock et al., 2020; Wu et al., 2020b; Dunne et al., 2020); the treatment for transport of gaseous tracers
and aerosols by advection and vertical convection; and the dry deposition and wet scavenging schemes
(Su et al., 2022; Digby et al., 2024).
Meteorological conditions, including temperature, precipitation and surface wind simulations, have
critical impacts on local aerosol concentrations. Temperature simulations over the eastern regions of
China by CMIP6 models are very close to the observed data (Yang et al., 2021). Over the western regions,
a notable warm bias over Xinjiang in most CMIP6 models (Zhang et al., 2022b) may contribute to higher
planetary boundary layer height (Yue et al., 2021) and stronger vertical mixing, partly explaining the
underestimation of OC and BC concentrations near the surface (Fig. 5); whereas the pronounced cold
bias over the Tibetan Plateau (Zhu and Yang, 2020) might contribute to overestimated near-surface
aerosol concentrations over there. Precipitation affects aerosol concentrations through wet scavenging;
and it is overestimated (wet bias) in CMIP6 models over North China and Northeast China but close to
observations over Central and South China (Yang et al., 2021). The model performance in precipitation
may partly explain the more severe underestimation of OC concentrations over North China than over
Central and South China. But the overestimation of BC over North China suggests that other factors
offset the influence of local wet bias. Over the western regions, most models exhibit wet bias, except
over northern Xinjiang where local temperature (warm bias) and precipitation (dry bias) have opposite
effects on near-surface aerosol concentrations. Furthermore, the overall underestimation of surface wind
speed over China in CMIP6 (Wu et al., 2020a) is conducive to the accumulation of near-surface aerosol
concentrations around the anthropogenic emission source regions, which may induce a negative
contribution to the underestimation of OC and BC concentrations.
**4.2 Sulfate, nitrate and ammonium**
This section evaluates the model performance of secondary inorganic aerosols (sulfate, nitrate, and
ammonium; SIOA). Sulfate aerosol in CMIP6 models is dependent on $SO_2$ emissions (the main sulfuric
acid precursor), chemical conversion of $SO_2$ to sulfate, and loss through wet scavenging (Wu et al., 2020b;
Tegen et al., 2019). Some models also explicitly simulate nitrate and ammonium aerosols using the
sulfate-nitrate-ammonia thermodynamic equilibrium. For instance, EC-Earth3-AerChem, GISS-E2-1-
MATRAX and GISS-E2-1-OMA use the Equilibrium Simplified Aerosol Model (EQSAM) (Metzger et
al., 2002; Bauer et al., 2020; van Noije et al., 2021), while GFDL-ESM4 treats ammonium and nitrate
aerosols with ISORROPIA (Fountoukis and Nenes, 2007; Paulot et al., 2016; Dunne et al., 2020).
The national average multi-year mean of observed sulfate concentrations reaches 14.6 μg m$^{-3}$, the second
largest value among the five PM$_{2.5}$ components (following OC). The observed sulfate concentrations
exceed 15 μg m$^{-3}$ over most of North China and eastern Sichuan, as well as cities over Xinjiang with
large population and petroleum industry (Fig. 5 g). The 14-model mean, whose national average is 9.3
μg m$^{-3}$, has the greatest underestimation over North China and Xinjiang (Fig. 5 h). The 14-model mean
agrees modestly well with the observations in spatial pattern (R = 0.57). Among the 14 models, the
national average model biases range from –66.1% (GISS-E2-1-OMA) to 24.5% (MRI-ESM2-0); and
five models better capture the observed spatial pattern with correlation coefficients exceeding 0.6 (Fig.
6). The cross-model discrepancy in sulfate (2 μg m$^{-3}$ in national average) is larger than those for the other
four components (0.4–0.9 μg m$^{-3}$), particularly over Central and South China (Fig. 5 i).
The national average multi-year mean of observational nitrate concentrations is 8.7 μg m$^{-3}$. The observed
spatial pattern of nitrate is similar to sulfate, with high values over North China, eastern Sichuan and
populous cities of Xinjiang (Fig. 5 j). Only four models (GFDL-ESM4, GISS-E2-1-OMA, GISS-E2-1-
MATRIX, and EC-Earth3-AerChem) include nitrate simulations. The 4-model mean has a national
average of 5.5 μg m$^{-3}$, with a NMB of –36.5%; but it captures the observed spatial pattern very well with
a correlation coefficient reaching 0.7. All the four models exhibit negative NMBs ranging from –41.4%
to –25.4%; they reproduce high values over the eastern regions but have underestimation over Xinjiang
(Fig. S7).
The observed multi-year mean ammonium concentrations have a national average value of 6.7 μg m$^{-3}$.
The observational values peak over North China (> 10 μg m$^{-3}$), particularly over the agricultural regions
from which ammonia emissions are the greatest (Fig. 5 m). Five models perform ammonium simulations.
The 5-model mean, with a national average of 3.4 μg m$^{-3}$, has negative and positive biases between –
12.2 and 1.5 μg m$^{-3}$ at different locations (Fig. 5 n). The 5-model mean captures the observed spatial
pattern of ammonium (R = 0.74) better than for other components (R = 0.39–0.70). The five models
exhibit varying performances in magnitude and spatial pattern. The NMBs range from –89.0% to –13.6%
across these models. Four models simulate the spatial patterns of ammonium well with high correlation
coefficients between 0.67 to 0.76, although the spatial agreement is poor for CESM2-WACCM (R =

359    0.21).

Emissions, meteorological conditions and chemical processes affect the formation and loss of secondary
inorganic aerosols. As explained in Sect. 4.1, the potentially overestimated CEDS emissions over China,
the cold bias over the Tibetan Plateau, and the dry bias over northern Xinjiang tend to overestimate
aerosol concentrations, which are in contrast with the negative model biases over the respective regions.
On the other hand, the warm bias over northern Xinjiang and the wet bias over North China and Northeast
China are in line with the underestimation of aerosol concentrations. Furthermore, the formation of
nitrate from nitric acid depends on the amount of residual ammonia left from the formation of ammonium
sulfate. Over the regions where ammonia is not sufficient to neutralize both nitric acid and sulfuric acid
(such as Shanxi and Shandong), decreased sulfate formation might promote nitrate formation with the
released ammonium (Zhai et al., 2019; Zhai et al., 2021). This partly explains why the underestimation
of nitrate simulations is less than sulfate over these regions.

**5 Discussion**

Over the eastern regions, the concentrations of total $PM_{2.5}$ and its five components are underestimated
by the 14 models in general. The slight underestimation of three models (GFDL-ESM4, MPI-ESM-1-2-
HAM, and MRI-ESM2-0) can be traced to positive biases in sulfate simulations partly offsetting the
negative biases in OC and BC. Over the western regions, most models underestimate the total $PM_{2.5}$
concentrations dominated by dust aerosols, whereas three models (GFDL-ESM4, NorESM2-LM, and
NorESM2-MM) produce overly high values over Xinjiang due to overestimated dust concentrations.
Meanwhile, all models underestimate the five $PM_{2.5}$ components over the west.
Figure 7 shows little difference between the maximum and average annual concentrations over 2000–
2014 for national mean $PM_{2.5}$ components simulated by individual models. Furthermore, we average
over all seasonal and annual observational records to compare with annual mean model results. A test
using the seasonal (annual) model results to match seasonal (annual) observational records shows very
similar comparison results (Fig. S8). These tests suggest that the model underestimation cannot be
attributed to imperfect temporal matching between models and observations or the potential mis-phase
(or variability) in models.
Among the five $PM_{2.5}$ components evaluated, absorbing aerosol (BC) and four scattering aerosols (OC,
sulfate, nitrate, and ammonium) have opposite direct radiative forcing at the top of atmosphere (TOA).
The underestimation of BC is less than for the other four scattering aerosols. If this difference persists in
the troposphere, the underestimated $PM_{2.5}$ components might cause an underestimation of negative
radiative forcing at TOA. The underestimation of BC and scatter aerosols might result in more solar
radiation reaching the ground (Chen et al., 2022; Tang et al., 2022). This is consistent with the
overestimation of maximum daily maximum temperature over the eastern regions (Zhu et al., 2020),
likely serving as a positive feedback between negative aerosol biases and overestimated surface
temperature.
The spatial biases in aerosols might also serve as an important limiting factor for the performance of

meteorological/climate simulations. The observed $PM_{2.5}$ and its five components are characterized by high concentrations over the east and low values over the west (except northern Xinjiang). In a few models, the large overestimation of $PM_{2.5}$ over Xinjiang of the west (dominated by dust) with underestimated $PM_{2.5}$ (dominated by anthropogenic aerosols) over the east might exert an incorrect west-east asymmetric climate forcing. The spatial pattern of resulting climate response might include cold-warm biases of surface temperature (cold bias over the west and warm bias over the east). The difference in the spatial pattern of model bias between BC and scattering aerosols might have additional impacts on the climate. Future work is needed to examine how the model errors in $PM_{2.5}$ and its components might affect climate simulations through aerosol-climate feedback.

**6 Summary**

In this study, we evaluate the performance of 14 CMIP6 ESMs in simulating total near-surface $PM_{2.5}$ and its five components over China during 2000–2014, and discuss the likely causes for model errors, and their climate implications. Our assessment helps to understand the capability of the current-generation models in the simulation of aerosols and aerosol-climate interactions, towards further improvement of climate predictions and projections. Our findings are summarized as follows:

(1) Twelve of the 14 CMIP6 models tend to underestimate the total $PM_{2.5}$ concentrations over China (NMB = –65.5% to –1.1%) and the other two models overestimate them (NMB = 17.0%–39.2%), as compared to a satellite-based dataset. The seven models that output total $PM_{2.5}$ concentrations exhibit underestimation between –47.9% and –3.3% over the eastern regions, although four of them capture the observed spatial pattern (R > 0.9). Over the western regions, four of these seven models reproduce the maximum center over southern Xinjiang. The seven models, for which we calculate the total $PM_{2.5}$ concentrations from outputted components, underestimate the observed $PM_{2.5}$ by –65.5% to –48.0% averaged over the country, in part due to missing nitrate and ammonium in the models.

(2) Over the eastern regions, all models simulate significant increasing trends of total $PM_{2.5}$ (0.32–1.14 $\mu g\ m^{-3}\ yr^{-1}$) over 2000–2014 that are close to satellite-based data (0.72 $\mu g\ m^{-3}\ yr^{-1}$). The models also capture the interannual variability of satellite $PM_{2.5}$ over Northeast China and North China. Over the western regions, 11 models simulate growing $PM_{2.5}$ concentrations at rates of 0.10–0.28 $\mu g\ m^{-3}\ yr^{-1}$, in

contrast to no significant trends in satellite data.
(3)   The 14-model mean captures the spatial pattern of observed OC modestly well (R = 0.51), but it
exhibits severe underestimation nationwide (NMB = –59.0%), with negative biases exceeding –50% in
11 models. The 14-model mean shows a poor capability in capturing the BC spatial pattern (R = 0.39),
and it also underestimates the BC observations (NMB = –27.2%). Two models exhibit positive biases in
BC, while the other 12 models exhibit negative biases.
(4)   Fourteen, four and five models output the sulfate, nitrate, and ammonium, respectively. The 14-
model mean of sulfate exhibits modest spatial correlation and bias (R = 0.57, NMB = –36.5%); and there
exist large discrepancies among these models, with biases ranging from –66.1% to 24.5%. The 4-model
mean of nitrate captures the spatial pattern well (R = 0.7), although it still underestimates concentrations
nationwide (NMB = –36.5%). The 5-model mean of ammonium has the best performance in reproducing
the spatial pattern (R = 0.74) but with a negative bias in magnitudes (NMB = –46.5%).
(5)   The overall underestimation of $PM_{2.5}$ and its components are associated with imperfectness in
emissions as input, modeled meteorology and chemistry. The underestimated $PM_{2.5}$ and its components
might cause an overall underestimated cooling effect at TOA and stronger warming at the surface in the
models. The model performance in spatial pattern differs between BC and scattering aerosols; and a few
models also exhibit strong positive biases over the west (associated with dust) but negative biases over
the east. Together, the errors in spatial pattern might have additional consequences for the modeled
climate. Further studies are warranted to quantify how model errors in the magnitude and spatial pattern
of aerosols affect the regional and global climate, for example, through the Regional Aerosol Model
Intercomparison Project (RAMIP) (Wilcox et al., 2022).
As a final note, those causes for aerosol underestimation may also affect ozone, and the underestimated
aerosol concentrations might also further affect the ozone simulation through radiative or heterogeneous
chemical processes (Jacob, 2000; Lin et al., 2012; Li et al., 2019). In addition, as CMIP6 models are also
used to study the health impacts of aerosols (Xu et al., 2022; Shim et al., 2021), the aerosol
underestimation needs to be corrected to allow a more reliable estimate of health consequences.

**Data availability**

CMIP6 data are available on the Earth System Grid Federation (ESGF) and can be freely downloaded via the website interface https://esgf-data.dkrz.de/search/cmip6-dkrz/ (last access: 8 September 2020, WCRP, 2020). Satellite-derived surface $PM_{2.5}$ concentration products can be accessed from the Washington University Atmospheric Composition Analysis Group website as version V4.CH.03 at https://sites.wustl.edu/acag/datasets/surface-pm2-5/. Observational data used in this paper are provided in the SI, with raw data available upon request to the corresponding author Jintai Lin (linjt@pku.edu.cn).

**Author contributions**

JL led the study. FR and JL designed the study, analyzed the results, and wrote the paper. CX provided the map data of four regions in China. JA collected observation data of $PM_{2.5}$ components from the literature. JW helped to analyze the evaluation results. RM, AD and MH provided satellite-derived data of total $PM_{2.5}$. ST performed UKESM1-0-LL and HadGEM3-GC31-LL simulations. NO performed MRI-ESM2-0 simulations. JZ performed BCC-ESM1 simulations. SB and KT performed GISS-E2-1-OMA and GISS-E2-1-MATRIX simulations. ØS performed NorESM2-LM and NorESM2-MM simulations. PN performed CNRM-ESM2-1 simulations. DN performed MPI-ESM1-2-HAM simulations. GS performed CESM2-WACCM simulations. TN and PS performed EC-Earth3-AerChem simulations. LH performed GFDL-ESM4 simulations. TT performed MIROC-ES2L simulations. All authors commented on the manuscript.

**Competing interests**

The authors declare that they have no conflict of interests.

**Financial support**

Jintai Lin and Fangxuan Ren have been supported by the National Natural Science Foundation of China (grant no. 42075175) and the second Tibetan Plateau Scientific Expedition and Research Program 525 (grant no. 2019QZKK0604). Naga Oshima has been supported by the Environment Research and Technology Development Fund (grant nos. JPMEERF20202003 and JPMEERF20232001) of the Environmental Restoration and Conservation Agency provided by Ministry of the Environment of Japan, the Arctic Challenge for Sustainability II (ArCS II, grant no. JPMXD1420318865), and the Global

Environmental Research Coordination System from Ministry of the Environment, Japan (grant no.
MLIT2253). David Neubauer has been supported by the European Union's Horizon 2020 research and
innovation programme project (FORCeS, grant no. 821205). Randall Martin has been supported by
NASA (grant no. 80NSSC21K0508).

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

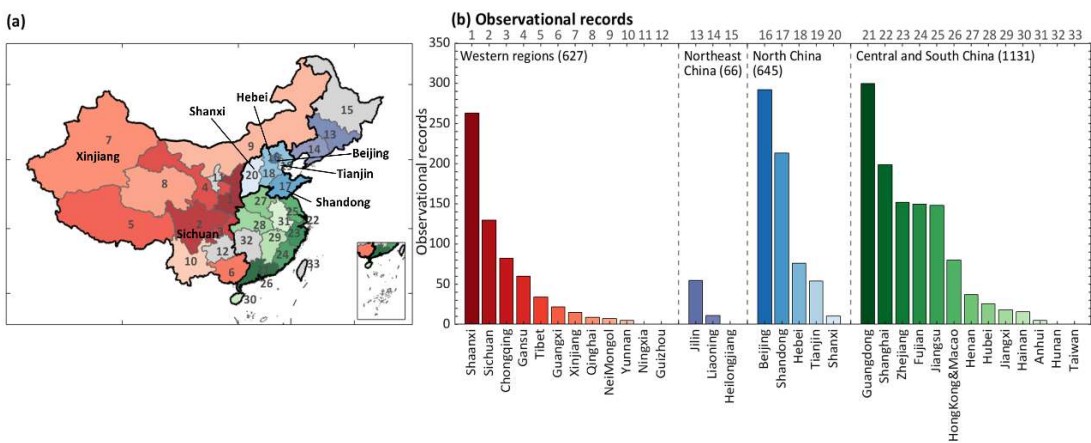


**Figure 1.** Observational records of PM$_{2.5}$ components during 2000–2014 collected from the literature. (a) The map
depicts individual provinces in four regions, including the western regions in red colors, Northeast China in purple,
North China in blue, and Central and South China in green. The provinces without observational records are in gray.
The number denotes each province. (b) Provincial observation records in China. The number in the upper x-axis and

the color in each bar match the province in (a).

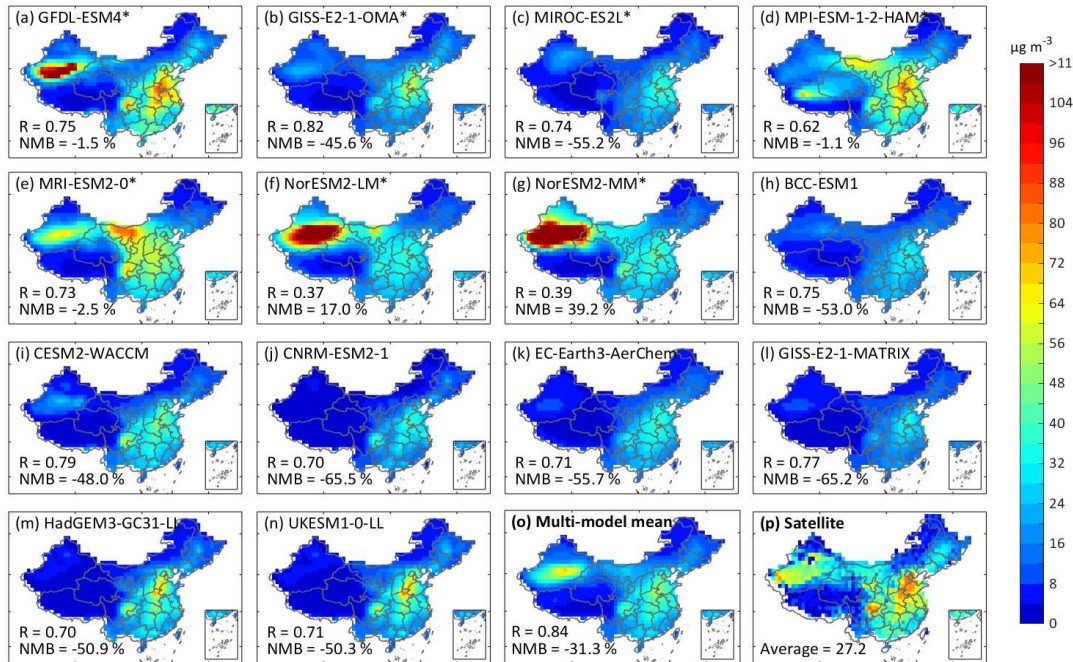

**Figure 2.** Multi-year mean annual average near-surface total PM$_{2.5}$ concentrations over China during 2000–2014.
(a-g) Model outputted PM$_{2.5}$ concentrations in seven models. (h-n) Calculated PM$_{2.5}$ concentrations in the other
seven models according to Eq. 1. (o) Multi-model mean. (p) Satellite-based total PM$_{2.5}$ dataset. R stands for spatial
correlation, and NMB stands for normalized mean bias.

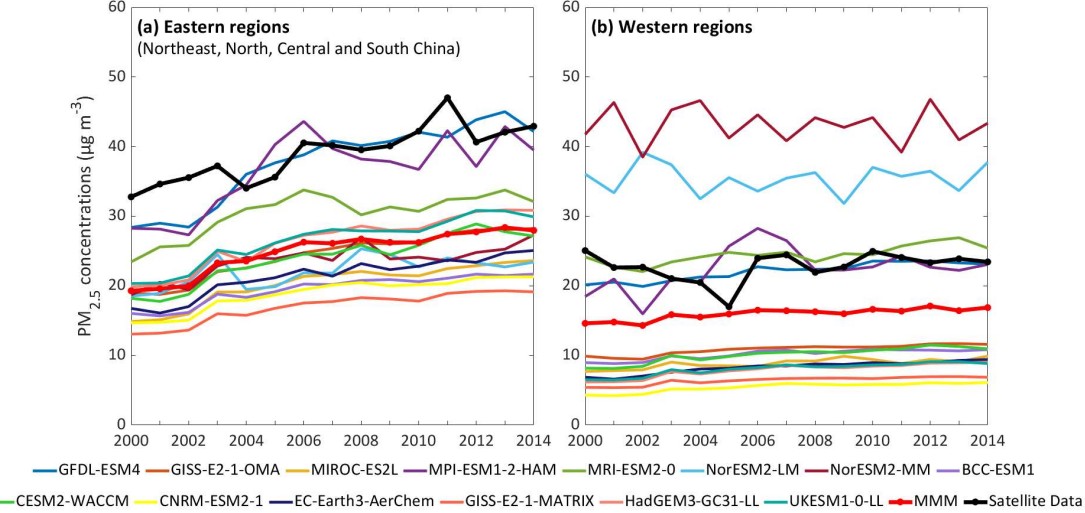

**Figure 3.** Time series of annual mean regional average total PM$_{2.5}$ concentrations. (a) Over the eastern regions
(including Northeast China, North China, and Central and South China). (b) Over the western regions. The bold
black lines denote satellite-based PM$_{2.5}$ concentrations, and the bold red lines denote multi-model mean (MMM)
concentrations.

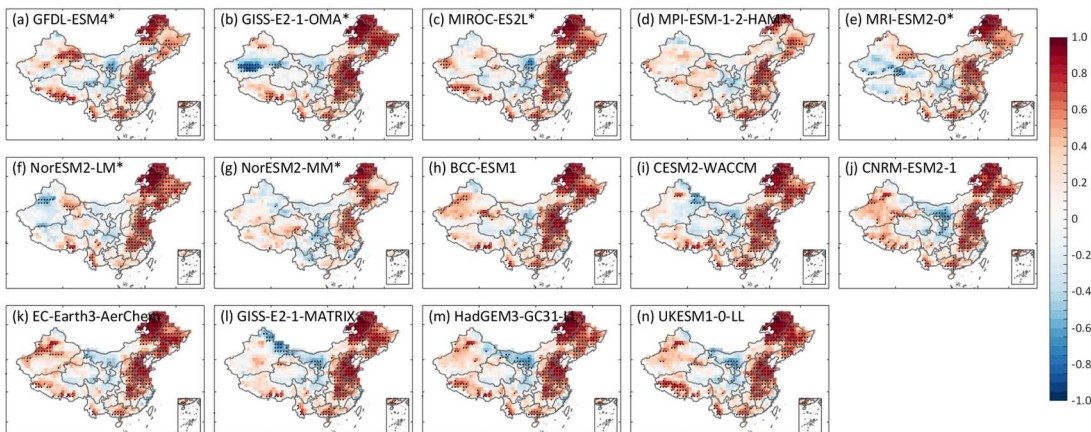

**Figure 4.** Spatial distribution of correlation coefficients between modeled and satellite-based data for interannual variations of annual mean total PM$_{2.5}$ concentrations during 2000–2014. Black dots indicate a significance level of 0.05.

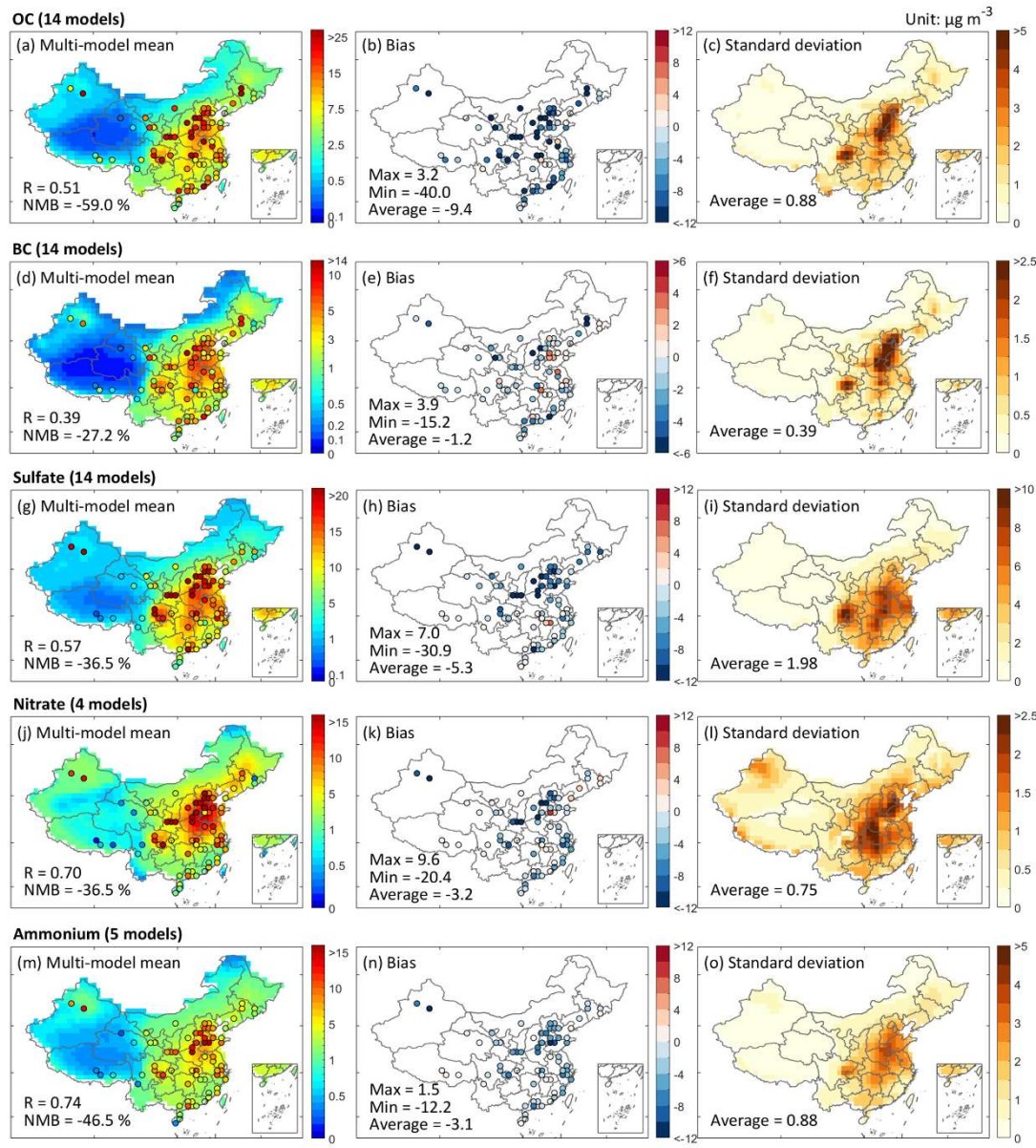

**Figure 5.** Spatial distribution of multi-year averages of modeled PM$_{2.5}$ components during 2000–2014. (First column) The multi-model mean PM$_{2.5}$ component concentrations, overlaid with average ground-based observations in filled circles. (Second column) The bias of multi-model mean concentrations. (Third column) The standard deviation of PM$_{2.5}$ component simulations among the CMIP6 models.

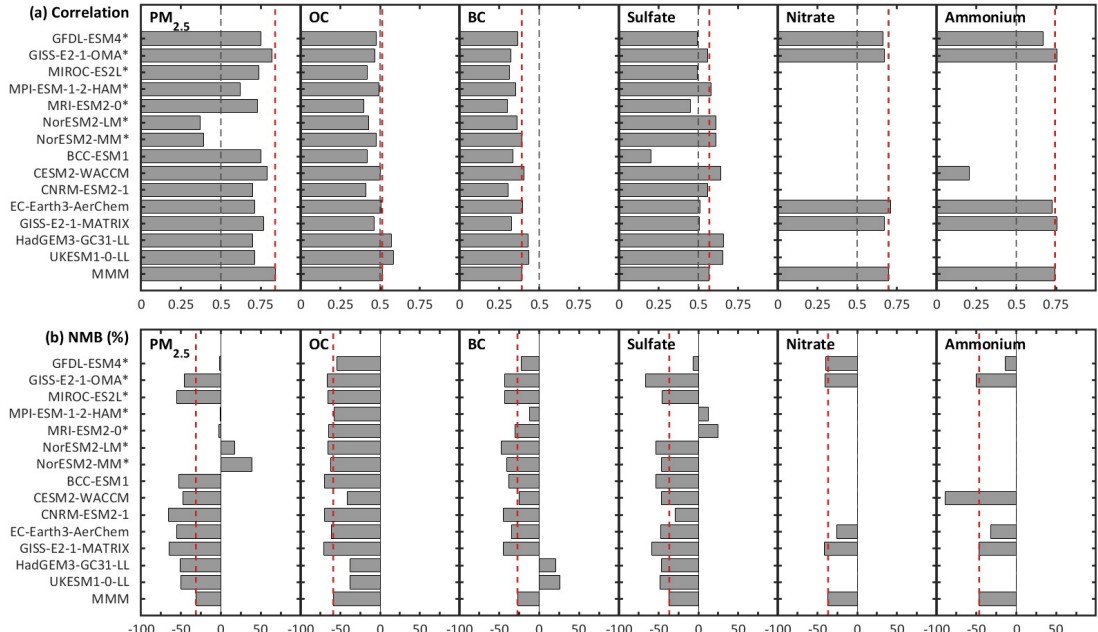

780

**Figure 6.** Multi-year mean spatial correlation and bias for PM$_{2.5}$ components over 2000–2014 for individual models.

Results for total PM$_{2.5}$ refer to the comparison against the satellite-based dataset, and those for components are

relative to the observations compiled from the literature. The red dotted lines denote multi-model mean (MMM).

The black dotted lines denote the spatial correlation coefficient value of 0.5.

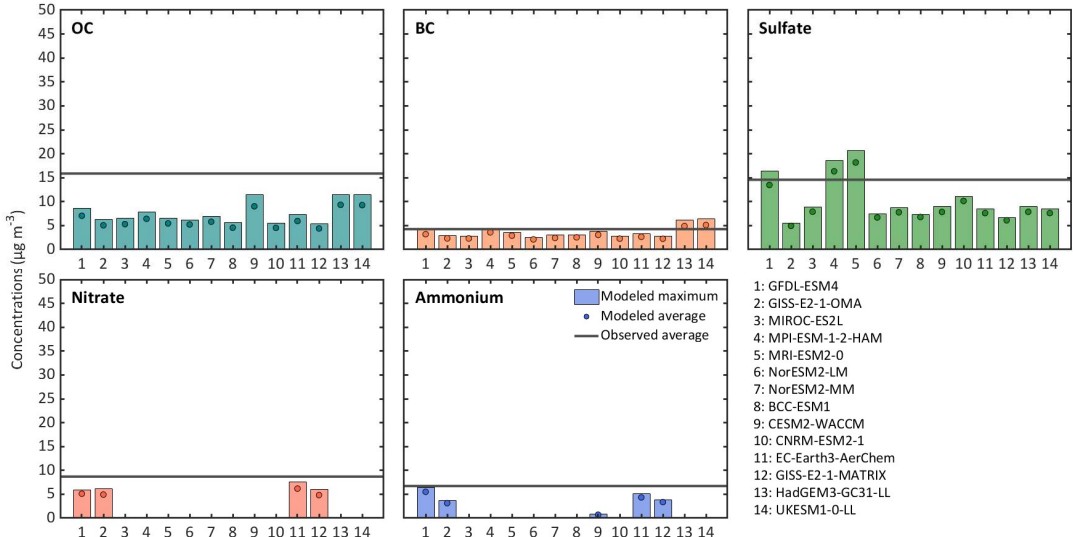

785

**Figure 7.** Maximum and average concentrations over 2000–2014 for simulated national mean PM$_{2.5}$ components

simulated by individual models. In each year, model values are sampled from grid cells with available observations.