# Peer review of "Evaluation of CMIP6 model simulations of PM2.5 and its"

_EGUsphere, 2023_

## Referee Comment (RC1)

Review of "**Evaluation of CMIP6 model simulations of PM$_{2.5}$ and its components over China**" by Ren et al.

This paper explores the performance of 14 CMIP6 models in simulating the spatial distribution, temporal variations, and components of PM$_{2.5}$ concentrations in China by comparing the models' historical run from 2000–2014 with satellite-based total PM$_{2.5}$ concentrations and ground-based PM$_{2.5}$ components data derived from the literature. It is found that PM$_{2.5}$ concentrations are generally underestimated, especially in eastern China. The concentrations of five individual components (OC, BC, sulfate, nitrate, and ammonium) are also largely underestimated. The potential causes of model biases and climate impacts on aerosol radiative forcing are also discussed. Overall, the paper is well-written, offering a thorough analysis and discussion that enhances our current understanding of the capabilities of the latest Earth system models. I have a few minor suggestions for the authors to consider.

1. Since the satellite-based PM$_{2.5}$ is one of the primary datasets used to validate CMIP6 output, it would be informative to include details about the dataset in section 2.2. This could involve specifying the aerosol species simulated by the GEOS-Chem model, discussing the accuracy of this dataset in comparison with PM$_{2.5}$ ground observations (if available), and providing information on the ground observations in China used in the dataset.

2. It may be helpful to explain why dust and sea salt concentrations are excluded from the PM$_{2.5}$ component analysis. For instance, are the ground observations available for these components?

3. In Section 3.2, is the positive trend in PM$_{2.5}$ concentrations in eastern China consistent with findings from previous studies? Do the emissions also show a positive trend in the same period?

4. In line 85, I'm not sure Sockol and Griswold (2017) examined PM$_{2.5}$ concentrations.

5. In line 214, none of the correlations in Fig. 2 are greater than 0.9.

6. In Fig. 2, what does NMB stand for? Normalized mean bias?

7. In Fig. 4, consider marking the area where correlations are statistically significant.

8. In Fig. 6, what do the dotted black lines denote?

---

## Author Comment (AC1)

Responses to Reviewer 1's comments

This paper explores the performance of 14 CMIP6 models in simulating the spatial distribution, temporal variations, and components of $PM_{2.5}$ concentrations in China by comparing the models' historical run from 2000–2014 with satellite-based total $PM_{2.5}$ concentrations and ground-based $PM_{2.5}$ components data derived from the literature. It is found that $PM_{2.5}$ concentrations are generally underestimated, especially in eastern China. The concentrations of five individual components (OC, BC, sulfate, nitrate, and ammonium) are also largely underestimated. The potential causes of model biases and climate impacts on aerosol radiative forcing are also discussed. Overall, the paper is well-written, offering a thorough analysis and discussion that enhances our current understanding of the capabilities of the latest Earth system models. I have a few minor suggestions for the authors to consider.

Reply: We thank a lot the Reviewer #1 for the comments. We have studied the comments carefully and tried to incorporate as many suggested changes as possible, which have greatly helped us in improving the manuscript. Our responses to the comments and suggestions are as follows. The original comments are in green while our replies are in black.

1. Since the satellite-based $PM_{2.5}$ is one of the primary datasets used to validate CMIP6 output, it would be informative to include details about the dataset in section 2.2. This could involve specifying the aerosol species simulated by the GEOS-Chem model, discussing the accuracy of this dataset in comparison with $PM_{2.5}$ ground observations (if available), and providing information on the ground observations in China used in the dataset.

Reply: As suggested, we have added the description in Lines 175-182 and cited it here:

"The GEOS-Chem aerosol simulations include primary and secondary carbonaceous aerosols, sulfate, nitrate, ammonium, mineral dust, and sea salt. The dataset provides the annual average $PM_{2.5}$ concentrations during the period 2000–2014 with a high spatial resolution of $0.01° × 0.01°$ (~1 × 1 $km^2$). The adjusted satellite-derived $PM_{2.5}$ concentrations over Asia are compared with surface $PM_{2.5}$ observations collected from the Global Burden of Disease (GBD) collaborators during the period 2008–2013 ($Mean_{satellite}$ = 61.5 μg m$^{-3}$ versus $Mean_{obs}$ = 59.1 μg m$^{-3}$) (van Donkelaar et al., 2016) and from the China National Environmental Monitoring Center (CNEMC) during the period 2015–2019 ($Mean_{satellite}$ = 45.9 μg m$^{-3}$ versus $Mean_{obs}$ = 43.4 μg m$^{-3}$) (van Donkelaar et al., 2021)."

2. It may be helpful to explain why dust and sea salt concentrations are excluded from the $PM_{2.5}$ component analysis. For instance, are the ground observations available for these components?

Reply: Thanks for pointing it out. As suggested, we have added the explanation in Lines 148-150 and cited it here:

"In evaluating $PM_{2.5}$ components (Sect. 4), the evaluation of dust and sea salt concentrations is excluded due to the lack of available ground-based observations. We compare OC, BC, sulfate, nitrate, and ammonium simulations with the observed data available for these components."

3. In Section 3.2, is the positive trend in $PM_{2.5}$ concentrations in eastern China consistent with

findings from previous studies? Do the emissions also show a positive trend in the same period?

Reply: The positive trend in PM$_{2.5}$ concentrations over eastern China is consistent with findings from previous studies. The aerosol optical depth (AOD) retrieved from the MODerate resolution Imaging Spectroradiometer (MODIS) over eastern China increased from 2000 to 2013 and decreased from 2015 to 2018 (de Leeuw et al., 2022). Geng et al. (2021) developed an air pollutant database named Tracking Air Pollution in China (TAP, http://tapdata.org.cn/) using information from monitor-, satellite-, and simulation-based sources. The TAP data also captures the PM$_{2.5}$ concentrations increasing rapidly before 2006 and dropping sharply after 2013.

CMIP6 emissions over eastern China also have a positive trend in the same period (Fig. R1). We have added the explanation in Lines 244-246 and cited it here:

"The positive trend of satellite data over the eastern regions is consistent with findings from previous studies of AOD and PM$_{2.5}$ (de Leeuw et al., 2022; Geng et al., 2021), as caused mainly by emission changes (Hoesly et al., 2018; Wang et al., 2022)."

[Figure]

**Figure R1.** Trends of CMIP6 emissions for BC, CO, NH$_3$, NMVOC, NO$_x$, OC, and SO$_2$ over eastern China from 2000 to 2014.

4. In line 85, I'm not sure Sockol and Griswold (2017) examined PM$_{2.5}$ concentrations.

Reply: Thanks for pointing it out. We have modified the expression in Lines 83-86 and cited it here:

"Several studies have evaluated total PM$_{2.5}$ simulations of CMIP models over China, using AOD data from satellite retrievals (Sockol and Small Griswold, 2017; Michou et al., 2020) and ground-based aerosol networks (Mortier et al., 2020)."

5. In line 214, none of the correlations in Fig. 2 are greater than 0.9.

Reply: The correlation coefficient mentioned in Line 214 is the spatial correlation between

simulations and satellite-based data over the eastern regions. We wanted to show that four models reproducing the spatial pattern over the eastern regions well with correlation coefficients greater than 0.9.

To make it clearer, we have added the spatial correlation coefficient values over the eastern and western regions into Table S2, respectively. We have also modified the expression in Lines 226-228 and cited it here:

"Nevertheless, the spatial pattern over the eastern regions is well simulated by four models (GFDL-ESM4, GISS-E2-1-OMA, MIROC-ES2L, and MPI-ESM-1-2-HAM) (R > 0.9, as shown in Table S2) with the maximum center over North China correctly reproduced."

**Table S2.** The specific values of $a_1$ and $a_2$ from Eq. 1. The average, trend, and spatial correlation coefficients of $PM_{2.5}$ concentrations over the eastern regions and western regions during 2000−2014.

| | Model | $a_1$ | $a_2$ | Eastern regions | | | Western regions | | |
|---|---|---|---|---|---|---|---|---|---|
| | | | | Average ($\mu g\ m^{-3}$) | Trend ($\mu g\ m^{-3}\ yr^{-1}$)[a] | Spatial Corr.[b] | Average ($\mu g\ m^{-3}$) | Trend ($\mu g\ m^{-3}\ yr^{-1}$) | Spatial Corr. |
| **Satellite-based** | | | | 39.0 | 0.72 | 1 | 22.7 | 0.06* | 1 |
| **Total PM$_{2.5}$ from Direct ESM output** | GFDL-ESM4 | | | 37.7 | 1.14 | 0.92 | 22.1 | 0.28 | 0.66 |
| | GISS-E2-1-OMA | | | 24.4 | 0.69 | 0.91 | 10.9 | 0.13 | 0.79 |
| | MIROC-ES2L | | | 20.3 | 0.49 | 0.90 | 8.9 | 0.13 | 0.59 |
| | MPI-ESM-1-2-HAM | | | 36.6 | 0.93 | 0.91 | 22.5 | 0.20* | 0.36 |
| | MRI-ESM2-0 | | | 30.4 | 0.57 | 0.83 | 24.5 | 0.24 | 0.71 |
| | NorESM2-LM | | | 22.1 | 0.32 | 0.87 | 35.5 | 0.03* | 0.49 |
| | NorESM2-MM | | | 23.6 | 0.40 | 0.90 | 43.1 | −0.10* | 0.53 |
| **Total PM$_{2.5}$ from Eq. 1** | BCC-ESM1 | | | 19.5 | 0.40 | 0.87 | 10.2 | 0.15 | 0.62 |
| | CESM2-WACCM | 0.25 | 0.1 | 24.0 | 0.73 | 0.92 | 10.1 | 0.22 | 0.67 |
| | CNRM-ESM2-1 | 0.02 | 0.25 | 18.9 | 0.42 | 0.90 | 5.5 | 0.11 | 0.51 |
| | EC-Earth3-AerChem | 0.25 | 0.1 | 21.4 | 0.56 | 0.91 | 8.3 | 0.18 | 0.53 |
| | GISS-E2-1-MATRIX | 0.25 | 0.1 | 17.0 | 0.43 | 0.92 | 6.4 | 0.10 | 0.67 |

| HadGEM3-GC31-LL | 0.27 | 0.35 | 26.5 | 0.80 | 0.89 | 7.9 | 0.18 | 0.50 |
| UKESM1-0-LL | 0.27 | 0.35 | 26.5 | 0.71 | 0.89 | 8.1 | 0.18 | 0.52 |

[a] Trends are estimated using the Theil-Sen Median method (Theil, 1950; Sen, 1968). Significant changes are identified using the non-parametric Mann-Kendall test (Kendall, 1938). * represents non-significant monotonous change at p = 0.05. [b] Spatial correlation coefficients between simulations and satellite-based data over the eastern and western regions are calculated. The spatial correlation coefficients of 14 models are at the 0.05 significance level.

6. In Fig. 2, what does NMB stand for? Normalized mean bias?

Reply: Thanks for pointing it out. NMB stands for normalized mean bias (i.e., NMB = (Mean$_{simulation}$ / Mean$_{observation}$ − 1) × 100%). Mean$_{simulation}$ and Mean$_{observation}$ are the spatial average of simulated and observed concentrations, respectively. We have added the definition of NMB in Lines 269-273 and Lines 764-765, and cited it here:

Lines 269-273: "The national average of the 14-model mean (6.5 µg m$^{-3}$, normalized mean bias (NMB) = –59.0%), which are spatially coincidently sampled with the ground-based observations (i.e., model values are obtained from grid cells with available observations), severely underestimates the observations, especially over parts of North China with the bias reaching –40 µg m$^{-3}$ (Fig. 5 b)."

Lines 766-767: "R stands for spatial correlation, and NMB stands for normalized mean bias."

7. In Fig. 4, consider marking the area where correlations are statistically significant.

Reply: As suggested, we have marked the regions where the correlation coefficient is at the 0.05 significance level.

[Figure]

**Figure 4.** Spatial distribution of correlation coefficients between modeled and satellite-based data for interannual variations of annual mean total PM$_{2.5}$ concentrations during 2000–2014. Black dots indicate a significance level of 0.05.

8. In Fig. 6, what do the dotted black lines denote?

Reply: The dotted black lines denote the spatial correlation coefficient value of 0.5. We plotted the dotted black lines as references to compare the correlations of 14 models more clearly. We have added the description of dotted black lines behind Fig. 6 (Line 784) and cited it here:

"The black dotted lines denote the spatial correlation coefficient value of 0.5."

**Reference**

de Leeuw, G., Fan, C., Li, Z., Dong, J., Li, Y., Ou, Y., and Zhu, S.: Spatiotemporal variation and provincial scale differences of the AOD across China during 2000–2021, Atmos. Pollut. Res., 13, 101359, https://doi.org/10.1016/j.apr.2022.101359, 2022.

Geng, G., Xiao, Q., Liu, S., Liu, X., Cheng, J., Zheng, Y., Xue, T., Tong, D., Zheng, B., Peng, Y., Huang, X., He, K., and Zhang, Q.: Tracking Air Pollution in China: Near Real-Time PM2.5 Retrievals from Multisource Data Fusion, Environ. Sci. Technol., 55, 12106-12115, https://doi.org/10.1021/acs.est.1c01863, 2021.

Hoesly, R. M., Smith, S. J., Feng, L., Klimont, Z., Janssens-Maenhout, G., Pitkanen, T., Seibert, J. J., Vu, L., Andres, R. J., Bolt, R. M., Bond, T. C., Dawidowski, L., Kholod, N., Kurokawa, J. I., Li, M., Liu, L., Lu, Z., Moura, M. C. P., O'Rourke, P. R., and Zhang, Q.: Historical (1750–2014) anthropogenic emissions of reactive gases and aerosols from the Community Emissions Data System (CEDS), Geosci. Model Dev., 11, 369-408, https://doi.org/10.5194/gmd-11-369-2018, 2018.

Kendall, M. G.: A new measure of rank correlation, Biometrika, 30, 81-93, https://doi.org/10.1093/biomet/30.1-2.81, 1938.

Michou, M., Nabat, P., Saint-Martin, D., Bock, J., Decharme, B., Mallet, M., Roehrig, R., Séférian, R., Sénési, S., and Voldoire, A.: Present-Day and Historical Aerosol and Ozone Characteristics in CNRM CMIP6 Simulations, J. Adv. Model. Earth Syst., 12, e2019MS001816, https://doi.org/10.1029/2019MS001816, 2020.

Mortier, A., Gliß, J., Schulz, M., Aas, W., Andrews, E., Bian, H., Chin, M., Ginoux, P., Hand, J., Holben, B., Zhang, H., Kipling, Z., Kirkevåg, A., Laj, P., Lurton, T., Myhre, G., Neubauer, D., Olivié, D., von Salzen, K., Skeie, R. B., Takemura, T., and Tilmes, S.: Evaluation of climate model aerosol trends with ground-based observations over the last 2 decades – an AeroCom and CMIP6 analysis, Atmos. Chem. Phys., 20, 13355-13378, https://doi.org/10.5194/acp-20-13355-2020, 2020.

Sen, P. K.: Estimates of the Regression Coefficient Based on Kendall's Tau, J. Am. Stat. Assoc., 63, 1379-1389, https://doi.org/10.1080/01621459.1968.10480934, 1968.

Sockol, A. and Small Griswold, J. D.: Intercomparison between CMIP5 model and MODIS satellite-retrieved data of aerosol optical depth, cloud fraction, and cloud-aerosol interactions, Earth Space Sci., 4, 485-505, https://doi.org/10.1002/2017EA000288, 2017.

Theil, H.: A Rank-Invariant Method of Linear and Polynomial Regression Analysis, Proc. R. Neth. Acad. Sci., 386–392,

van Donkelaar, A., Martin, R. V., Brauer, M., Hsu, N. C., Kahn, R. A., Levy, R. C., Lyapustin, A., Sayer, A. M., and Winker, D. M.: Global Estimates of Fine Particulate Matter using a Combined Geophysical-Statistical Method with Information from Satellites, Models, and Monitors, Environ. Sci. Technol., 50, 3762-3772, https://doi.org/10.1021/acs.est.5b05833, 2016.

van Donkelaar, A., Hammer, M. S., Bindle, L., Brauer, M., Brook, J. R., Garay, M. J., Hsu, N. C., Kalashnikova, O. V., Kahn, R. A., Lee, C., Levy, R. C., Lyapustin, A., Sayer, A. M., and Martin, R. V.: Monthly Global Estimates of Fine Particulate Matter and Their Uncertainty, Environ. Sci. Technol., 55, 15287-15300, https://doi.org/10.1021/acs.est.1c05309, 2021.

Wang, C., Wang, Z., Lei, Y., Zhang, H., Che, H., and Zhang, X.: Differences in East Asian summer monsoon responses to Asian aerosol forcing under different emission inventories, Advances in Climate Change Research, 13, 309-322, https://doi.org/10.1016/j.accre.2022.02.008, 2022.

---

## Author Comment (AC2)

This manuscript presented a thorough and fundamental evaluation of the CMIP6 model simulations of $PM_{2.5}$ over China. CMIP6 simulations are widely applied for climate related studies and it is necessary to fully understand model uncertainty as aerosol plays an important role in the climate system. Yet the performance of global model for predicting surface $PM_{2.5}$ concentrations has been largely ignored as it was considered as a special task for chemical transport models, especially at regional scale. However, in recent decade many studies reported the significance of interactions between air pollution and climate, thus it is important to reveal how the CMIP6 simulations can reproduce surface $PM_{2.5}$ as well. The manuscript is well organized with clear description of model and observational data employed. It provides a thorough discussion of the results and origins of uncertainties with solid method. Therefore, I would recommend it to be accepted with minor revisions if the following comments could be properly addressed.

Reply: We thank a lot the Reviewer #2 for the comments. We have studied the comments carefully and tried to incorporate as many suggested changes as possible, which have greatly helped us in improving the manuscript. Our responses to the comments and suggestions are as follows. The original comments are in green while our replies are in black.

1. line158: It would be helpful to show the comparison between direct summary of all fine aerosol species (e.g., $PM_{2.5}$ = sulfate + oa + nitrate + ammonium + bc + fine dust + fine sslt + bc) and the value of this equation, for the 4 models which provide nitrate. This would help to demonstrate the accuracy of the equation.

Reply: Thanks for the suggestion. We have added the comparison of $PM_{2.5}$ concentrations between the value of Eq. 1 and the value including all fine aerosol species, for 4 models providing nitrate and ammonium. We have also calculated the proportion of nitrate and ammonium to total $PM_{2.5}$, respectively. The result is shown in Table S3. We have added the following descriptions in Lines 233-235 and cited it here:

"The negative biases are in part because nitrate and ammonium are not included. About 15.1–20.6% and 11.4–14.6% of $PM_{2.5}$ are nitrate and ammonium in the models that do contain them, as shown in Table S3."

**Table S3.** Multi-year averages of $PM_{2.5}$ concentrations including five aerosol species (Eq. 1) and all fine aerosol species from 4 models providing nitrate and ammonium simulations.

| Model | $PM_{2.5}$ according to Eq. 1 ($\mu g\ m^{-3}$) | $PM_{2.5}$ including all fine aerosol species ($\mu g\ m^{-3}$)[a] | Nitrate proportion[b] | Ammonium proportion |
|---|---|---|---|---|
| EC-Earth3-AerChem | 12.1 | 18.7 | 20.6% | 14.3% |
| GFDL-ESM4 | 13.0 | 18.5 | 15.1% | 14.6% |
| GISS-E2-1-OMA | 10.0 | 14.1 | 17.6% | 11.4% |

| GISS-E2-1-MATRIX | 9.5 | 13.8 | 17.5% | 13.2% |
| --- | --- | --- | --- | --- |

[a] represents that $PM_{2.5} = OA + BC + SO_4^{2-} + 0.25SSLT + 0.1DST + NO_3^- + NH_4^+$. [b] represents that the proportion of nitrate to $PM_{2.5}$ including all fine aerosol species.

2. line171: It would be helpful to show comparison between satellite product and model results for AOD at monthly scale to reveal the performance of model in simulating seasonal variations of aerosol over China. This may provide more indications of model uncertainty, such as dust may dominate in spring and OA may dominate in summer.

Reply: Evaluating the performance of seasonal simulations over China using AOD data is another interesting work, as Li et al. (2021) has done. Li et al. (2021) finds that CMIP6 models fail to capture the seasonal north-south shift of AOD maximum centers. The maximum centers of AOD in spring over South China and in summer over North China are underestimated due to the underestimation of organic aerosol (OA) AOD and sulfate AOD respectively. We have also mentioned it in Lines 92-94 in the introduction and cited it here:

"Nonetheless, the CMIP6 models fail to capture the seasonal north-south shift of AOD maximum center over China during 2000–2014 (Li et al., 2021) and the observed dipole pattern of AOD trends between China and India during 2006–2014 (Wang et al., 2021b)."

3. line194: Not sure what is "effect of interannual variability", please make it clear.

Reply: Apologies for any confusion. In sections 4.1 and 4.2, we compared multi-year mean simulations with multi-year mean observations in grid cells with available observed data. In the same grid cell, the observations are not continuous in time while simulations are continuous from 2000 to 2014. The potential problem is that the incomplete temporal match between models and observations exists due to the interannual variability. Therefore, we calculated the maximum values of annual mean simulations over 2000–2014 from grid cells with available observed data to test whether the overall underestimation of models is related to the incomplete match (in year) between models and observations. We find that there is a small difference between the average and maximum of annual mean values over 2000–2014 from these grid cells in the models. We have modified the expression to make it clear in Lines 204-208 and cited it here:

"To consider the effect of interannual variability (caused by incomplete temporal match in data availability between models and observations), we compute for each CMIP6 model the average and maximum of annual mean values during 2000–2014 from all grid cells with available observational data, and then compare with the multi-year averaged observations from these grid cells."

4. line201: A figure similar to Fig.4 but for bias between model and satellite-based product would be helpful to reveal the difference more clear.

Reply: As suggested, we have added the bias between simulations and satellite-based data as Figure S2.

[Figure]

**Figure S2.** Spatial distribution of bias in the multi-year average of simulate-based PM$_{2.5}$ concentrations during 2000–2014 for each model.

5. line209: It would be helpful to provide a brief discussion of why certain specific model show better performance than others.

Reply: As suggested, we have added a brief discussion about this problem in Lines 220-224 and cited it here:

"Among the seven models that directly output total PM$_{2.5}$ concentrations (Fig. 2 a-g), GFDL-ESM4 and MPI-ESM-1-2-HAM show similar patterns and magnitudes to satellite data with small national average biases (–1.5% and –1.1%, respectively) because of better performance in BC, sulfate, and ammonium simulations (Fig. S4-S7), which are related to the aerosol-chemistry-climate schemes within CMIP6 models (Turnock et al., 2020)."

6. line238: Can you add a brief discussion of possible causes for the decline over 2000-2005 in satellite data? Is it an observational fact or satellite bias, if it is a fact, then why model cannot reproduce it?

Reply: As suggested, we have added the explanation in Lines 254-259 and cited it here:

"There is a notable decline over 2000–2005 in satellite data (–1.12 µg m$^{-3}$ yr$^{-1}$, at the significance level of 0.1), consistent with the previous studies that use dust aerosol optical depth (DOD) and ground-based observations of dust storm (Wang et al., 2021a; Song et al., 2016). However, the dramatic drop is not captured by any model, reflecting large uncertainties and inter-model diversities in dust simulations stemming from many factors such as the driving mechanisms, dust particle size, and model structural differences (Zhao et al., 2022)."

7. line250: Not sure what is "spatially coincidently sampled", please make it clear.

Reply: Apologies for any confusion. What we wanted to express is that model values are obtained from grid cells with available observations. We calculated the national average of model values from these grid cells. We have added the explanation in Lines 269-273 and cited it here:

"The national average of the 14-model mean (6.5 µg m$^{-3}$, normalized mean bias (NMB) = –59.0%),

which is spatially coincidently sampled with the ground-based observations (i.e., model values are obtained from grid cells with available observations), severely underestimates the observations, especially over parts of North China with the bias reaching $-40$ μg m$^{-3}$ (Fig. 5 b)."

8. line255: It would be helpful to add a brief discussion to explain why model difference peaks at these regions.

Reply: As suggested, we have added the explanation in Lines 296-304 and cited it here:

"The inter-model discrepancies of OC and BC peak over North China and eastern Sichuan (Fig. 5 c). The large absolute discrepancies are in part due to the higher air pollutant concentrations in these regions. Furthermore, many differences exist among CMIP6 models in PM$_{2.5}$ component simulations, including the representation of aerosol size distribution; the simplification of chemical processes with photolytic, kinetic and heterogeneous reactions (e.g., 33 photolytic reactions in BCC-ESM1 but 43 in GFDL-ESM4) (Turnock et al., 2020; Wu et al., 2020; Dunne et al., 2020); the treatment for transport of gaseous tracers and aerosols by advection and vertical convection; and the dry deposition and wet scavenging schemes (Su et al., 2022; Digby et al., 2024). "

9. line259: Is it because of spatial distribution pattern of CEDS emission?

Reply: We have calculated the spatial correlation coefficients between three variables (i.e., CMIP6 emissions, multi-model mean concentrations, and observed concentrations). For OC and BC, the modeled concentrations are consistent with emissions (R > 0.85), while the consistency is lower between emissions and observed concentrations (R = 0.42–0.55) (Fig. S3). Thus the spatial distribution of emissions plays a major role in the model performance of carboneous aerosol simulations. In contrast, for sulfate and nitrate, the correlation coefficients are modest between emissions and simulated concentrations (R = 0.6) and between emissions and observed concentrations (R = 0.49–0.51), indicating a key role of chemical processes. We have added the expression in Lines 286-288 and cited it here:

"The spatial distributions of carbonaceous aerosol concentrations are mainly influenced by CEDS emissions used in models, with their spatial correlation coefficients greater than 0.85 (Fig. S3)."

[Figure]

**Figure S3.** Multi-year average of CMIP6 emissions (a-e), multi-model mean concentrations (f-j), and observed concentrations (k-o) of air pollutants over 2000–2014. R (e, m), R (e, o), and R (m, o) denote the spatial correlation coefficients between CMIP6 emissions and multi-model mean concentrations, between CMIP6 emissions and observed concentrations, and between multi-model mean concentrations and observed concentrations.

**10. line269: How much was CEDS emission data, as compared to which dataset?**

Reply: Fan et al. (2022) has compared the CEDS inventory for CMIP6 models with a country-level inventory (i.e., MEIC v1.3) over China. The BC, POM, $SO_2$, $NH_3$ emissions in CEDS are 26.3%, 10.5%, 31.3%, 3.8% higher than those in MEIC during 2006–2015 respectively, while $NO_x$ emission in CEDS is 21.7% lower than that in MEIC during the same period. We have added the expression in Lines 290-292 and cited it here:

"For China, the CEDS emission data (ver. 2016-07-26) used in CMIP6 historical simulations are about 3.8–31.3% higher than those in MEIC inventory except for $NO_x$ emissions (–21.7%) (Fan et al., 2022)."

**11. line272: BC is primary**

Reply: Apologies for any confusion. We have modified the expression in Lines 293-296 and cited it here:

"The model inadequacies in chemical processes (e.g., using simplified aerosols and chemistry schemes, which tends to underestimate aerosol formation (Turnock et al., 2020)) might lead to underestimated secondary organic aerosols (SOA, as a component of OC), especially over Central and South China (Chen et al., 2016)."

**12. line272: Do all models have warm bias over Xinjiang? In addition, some observation sites provide PBL measurements as well, so why not perform evaluation of simulated PBL directly?**

Reply: Most models have a warm bias over Xinjiang during the historical periods (2000–2014), according to the previous studies (Zhang et al., 2022; Fan et al., 2020). For example, Zhang et al. (2022) finds that climatological annual surface air temperature simulation over Xinjiang during

1995–2014 as obtained from the median and arithmetic mean of 42 CMIP6 models (including 14 models used in our study) are higher than the observations. For PBL evaluation, Yue et al. (2021) finds that the magnitude of PBL height in nine CMIP6 models is generally overestimated compared with the data from the fifth generation of the European Centre for Medium-Range Weather Forecasts (ECMWF) atmospheric reanalysis of the global climate (ERA5; Copernicus Climate Change Service 2017). We have added the PBL evaluation in Line 307-312 and cited it here:

"Over the western regions, a notable warm bias over Xinjiang in most CMIP6 models (Zhang et al., 2022) may contribute to higher planetary boundary layer height (Yue et al., 2021) and stronger vertical mixing, partly explaining the underestimation of OC and BC concentrations near the surface (Fig. 5); whereas the pronounced cold bias over the Tibetan Plateau (Zhu and Yang, 2020) might contribute to overestimated near-surface aerosol concentrations over there."

13. line292: It would be helpful to briefly explain how 14 models simulate sulfate formation chemistry.

Reply: As suggested, we have added the explanation in Lines 324-331 and cited it here:

"This section evaluates the model performance of secondary inorganic aerosols (sulfate, nitrate, and ammonium; SIOA). Sulfate aerosol in CMIP6 models is dependent on $SO_2$ emissions (the main sulfuric acid precursor), chemical conversion of $SO_2$ to sulfate, and loss through wet scavenging (Wu et al., 2020; Tegen et al., 2019). Some models also explicitly simulate nitrate and ammonium aerosols using the sulfate-nitrate-ammonia thermodynamic equilibrium. For instance, EC-Earth3-AerChem, GISS-E2-1-MATRAX and GISS-E2-1-OMA use the Equilibrium Simplified Aerosol Model (EQSAM) (Metzger et al., 2002; Bauer et al., 2020; van Noije et al., 2021), while GFDL-ESM4 treats ammonium and nitrate aerosols with ISORROPIA (Fountoukis and Nenes, 2007; Paulot et al., 2016; Dunne et al., 2020)."

14. line344: Not sure what is "seasonal model results to match seasonal observational", do you mean conduct model evaluation at seasonal scale, so all models and observations are averaged seasonally?

Reply: Apologies for any confusion. We do not focus on the model evaluation at the seasonal scale. In section 4, we calculated the multi-year average of model values at each grid cell with available observations, and compared it with the mean of all available observations at that grid cell. However, the observations do not have the same number of records in each season. For example, for a given grid cell, if the number of observed records in winter are more than the number in other seasons, the multi-year average of observations would have a higher weight from winter, whereas the modeled multi-year average has the same weight from each season. To evaluate how this treatment affects our model evaluation of multi-year means, we conducted tests to match model values with observations in the same season (e.g., if there are 50 observed records in winter, then we take multi-year winter mean values by 50 times from model simulations); this is described as the "seasonal model results to match seasonal observational records". We find that different matching methods cannot change the main conclusions.

15. line406: I would recommend to mention it as those causes for aerosol underestimation may also affect O3.

Reply: Changed as suggested. We have added the explanation in Lines 444-446 and cited it here:

"Those causes for aerosol underestimation may also affect ozone, and the underestimated aerosol concentrations might further affect the ozone simulation through radiative or heterogeneous chemical processes."

[revised manuscript text omitted]